

# Characterization of Marine Heat Waves in the IBI Region in 2022

Lluís Castrillo-Acuña[1], Axel Alonso-Valle[1,2], Álvaro de Pascual-Collar[1]

[1]Nologin Oceanic Weather Systems, Paseo de la Castellana 216, Floor 8th, Office 811, Madrid, 28046, Spain
[2]Earth Physics and Astrophysics Department, Complutense University of Madrid, Madrid, 28040, Spain

*Correspondence to*: L. Castrillo Acuña (lluis.castrillo@nologin.es)

**Abstract**

Marine heat waves (MHWs) are defined as prolonged periods of anomalously high sea surface temperatures. These events have a profound impact on marine ecosystems resulting in ecological and economic impacts such as coral bleaching, reduced surface chlorophyll due to increased surface layer stratification, mass mortality of marine invertebrates due to heat stress,

rapid species' migrations, fishery closures or quota changes, among others.

This research focuses on the study of the MHWs that occurred in the IBI region during the year 2022, assessing their climatologic properties, the mean values for the year 2022 and discretizing the events in four subregions representative of the entire domain. Satellite derived sea surface temperature data was used to detect and characterise the events, revealing that in some areas the year 2022 showed up peak anomaly values of (i) 10 MHWs events, (ii) 100 days of mean durations and (iii)

190 total days of MHW, above normal. Through observational and modelling data, the discrete events located in the Bay of Biscay were also examined in the subsurface layers, demonstrating a strong seasonal modulation and heat diffusion through deeper layers. Where cold season events reach higher MHW mean depth values and subsurface positive anomalies of temperature can remain during weeks once a MHW has ended.

## 1 Introduction

Marine heat waves (MHWs) are a physical process which perform extreme temperatures, at least, on the ocean surface. As they are known to be related with multiple drastic alterations in marine ecosystems and services (Holbrook et al., 2020; Smale et al., 2019), and due to the recently observed ocean surface warming of 0.88°C in the last decade (IPCC, 2023), the scientific community has shown a growing interest in this topic (Hobday et al., 2018).

In this contribution, an analysis of the MHWs in the IBI (Iberia-Biscay-Ireland) domain during the year 2022 is performed.

The IBI region is one of the areas handled by the Monitoring Forecasting Centers of the Copernicus Marine Service, located in the Northeastern Atlantic Ocean between the Canary Archipelago at south, and Great Britain and Ireland at north (Figure 1). This region clusters multiple dynamical systems, such us upwelling areas, open waters, straits, and bays, and it is hence a region characterized by a remarkable range of physical processes at various spatial and temporal scales (Sotillo et al., 2015).



To detect and analyse MHWs, the standard method of Hobday et al. (2016) is used, defining a MHW as a discrete event that
lasts for at least five consecutive days exhibiting temperatures warmer than the 90[th] percentile of the climatological
distribution. This method has been widely used and hence, an important number of comparable MHW studies around the
world have been published. Unfortunately, there is an unsolved issue regarding the Hobday et al. (2016) method; how to deal
with sea surface temperature (SST) trends and MHWs' detection. Different authors have assessed this issue, but a consensus
has not been reached yet. Through the use of synthetic SST time series and sensitivity experiments, Schlegel et al. (2019)
demonstrated that SST long term linear trends can have a much greater effect on the trend of MHW properties than the
length of the series or even the presence of missing data. Furthermore, the global assessment of Oliver et al. (2018) shows
that just the SST trend may explain the MHW trends in an 80%, 59% and 53% of the ocean surface for the frequency,
intensity, and duration, respectively.

Considering the results of the recent MHW global assessments, it is expected for such events to increase in their frequency
and duration during the next years in most parts of the world (Oliver et al. 2018; Yao et al. 2022; Collins et al. 2019, Fox-
Kemper et al. 2021). These predictions also include the IBI domain, a region characterised by Yao et al. (2022) as presenting
from 1982 to 2020 MHWs with an intensity mean close to 1ºC and 15 to 30 MHW days per year, approximately. A wide
range of physical processes can be pointed out as drivers of the occurrence of MHWs depending on the sub-regions assessed.
Specifically, our study area covers the Canary basin, the Iberian Peninsula, the Bay of Biscay and the Celtic Sea (Figure 1).

Canary and Iberian MHWs are mostly linked to processes of atmospheric blocking, the negative phase of the North Atlantic
Oscillation (NAO), the regional air-sea coupling, the regional changes of wave stress and the jet stream position, local
advective processes, and to air-sea heat fluxes (Holbrook et al. 2019; Varela et al. 2021). In a rare instance, the influence of
ENSO has also been observed in a record-breaking event recorded in the area (Hu et al. 2011).

In the case of the Bay of Biscay and the Celtic Sea the main interannual drivers of MHWs are the NAO and the East Atlantic
pattern (EA) (Izquierdo et al. 2022; Simon et al. 2023), while also other processes such as the inflow from the English
Channel and the strength of the tidal currents play a key role on the regional changes of the SST (Cornes et al. 2023).

In this research we aim to characterize the year 2022 regarding the MHWs in the IBI domain, considering not only the
annual mean values but also the 2022 discrete events in four different sub-regions representative of the domain. Also, we
shed light on the first steps of learning how MHWs behave under the surface by using Copernicus products.





## 2 Data and methods

In the present work several Copernicus Marine products (described in Table 1) have been used to provide a description of the MHWs which occurred in the IBI region during the year 2022. The diversity of products used is due to our leverage of their different strengths in the detection and description of MHWs.

### 2.1 Data

To detect the MHW events, we used the ESA SST CCI and C3S global Sea Surface Temperature Reprocessed product (GLO-REP, Table 1, product ref. 1), which is a homogenous level 4 analysis. This dataset provides daily gridded gap-free SST data from the 1st of September 1981 to the 30th of September of 2022 at 0.05deg. x 0.05deg. of spatial resolution. The input data of the system derives from three different satellite sensors, the ATSRs, the SLSTR and the AVHRR (Merchant et al., 2019), and it is processed through the Operational Sea Surface Temperature and Sea Ice Analysis (OSTIA) system developed by the UK's Met office (Good et al., 2020). The availability of gridded data for this product has enabled: (i) the generation of a reference climatology and seasonal threshold to detect MHWs, and (ii) the compilation of a catalogue of MHWs that have impacted the study area during 2022.

Once a specific event was located in space and time, we observed how some of these events behaved under the surface. To achieve this goal, we used sea water temperature data from the ocean surface down to 350 meters of depth from both in situ observations and numerical models. Thus, we examined specific events with *in situ* data, and also estimated their development during all the MWH days through numerical modelling data which has no spatial or temporal limitations.

Argo is the collective name of a global array of 3,000 automated free-drifting profiling floats that measure sea water temperature and salinity in the upper ocean as well as, in some cases, bio-geo parameters such as oxygen or chlorophyll concentration. All collected data is freely available by the international Argo project and the national programs that contribute to it (Argo 2019). The specific Copernicus Marine Environment Monitoring Service (CMEMS) product that we used is the Atlantic Iberian Biscay Irish Ocean- In-Situ Near Real Time Observations (hereafter ARGO product, Table 1, product ref. 2), which compiles level 2 processed in situ near real-time data from Argo floats and other observational sources in the IBI region since the 1st of January 1990 to the current day. It is hourly updated and distributed by the Copernicus Marine In Situ Thematic Assembly Centre (In Situ TAC) within 24-48 hours from acquisition. The ARGO observations consist of instantaneous values, quality-controlled and irregularly distributed in time and space, as a result of the diverse modes of operation, problems with the sensors and drifting movement of the buoys.

With the aim of acquiring data that allows a more detailed study at a regular daily scale, two three-dimensional, gridded and gap-free CMEMS datasets from numerical models have also been used, both run and provided by the IBI Monitoring and Forecasting Center. The first one is the Atlantic-Iberian Biscay Irish- Ocean Physics Analysis and Forecast (IBI-NRT, Table



85 1, product ref. 3), a product with a spatial resolution of 0.028deg. x 0.028deg. and 50 depth levels down to 5,728 meters. It provides best estimates with level 4 processing of different physical variables for the last two years, as well as a forecast with a 5-day horizon, updated daily. Secondly, we used the Atlantic-Iberian Biscay Irish- Ocean Physics Reanalysis (IBI-REA, Table 1, product ref. 4), which extends from the 1st of January 1993 to the 28th of December 2021. It has a spatial resolution of 0.083deg. x 0.083deg. with the same vertical levels as IBI-NRT, and a time resolution that ranges from hourly to yearly.

90 Observational data assimilated for the reanalysis include altimeter measurements, in situ temperature and salinity vertical profiles, and satellite sea surface temperature. For the purposes of this study, we extracted daily averaged values of potential temperature (θ) in the water column from 2005 to 2021 for IBI-REA, and the year 2022 for IBI-NRT. By such, we obtained a dataset to use as a mean reference (IBI-REA) and another one to assess the year 2022 (IBI-NRT) deep inside the ocean.

## 2.2 Methods

### 2.2.1 Surface MHW assessment

The study and detection of the MHWs was computed through the standard definition of Hobday et al. (2016) applied to the GLO-REP product from January 1982 to September 2022. We chose the usual parameters in order to obtain results comparable to those of similar studies on this topic: a minimum duration of 5 days to consider a MHW, a maximum gap 100 tolerance of 2 days between two events, the threshold was calculated through the 90$^{th}$ percentile, and the climatology and threshold computed for all the period were smoothed out using a moving window of 31 days.

Among the set of parameters available to characterise the MHW we selected the ones that we understand as fundamental to evaluate the state of MHWs in the IBI domain during 2022: the frequency of the events, the duration, the maximum intensity point relative to the climatology and the absolute value, and the cumulative intensity, which can be assessed as the total 105 energy of an event.

Regarding the possible presence of linear trends in SST, in this research we did not apply any kind of trend assessment nor a detrending method due to the lack of any standard procedure.

For a deeper analysis of MHWs in the region, we defined four subregions to be representative of the different oceanographic systems in our study area and performed a spatial average to assess them. According to this criterion, the selected subregions 110 were the Continental shelf near to British Islands and English Channel (CEL), the offshore region of the Gulf of Biscay (BSC), the upwelling region next to the coast of the Iberian Peninsula (IBE) and the Azores and Canary Islands basin (CAN) (Figure 1). In this manner, we were able to analyse the discrete events in 2022 and the record-breaking ones for all the years as a reference for each sub-domain.





### 2.2.2 Subsurface MHW assessment

The Argo floats network is used to assess specific events from the ocean surface down to a maximum depth of 350 meters. With the aim of computing a temperature anomaly or deviation profile which represents a single event, we first converted pressure into depth by using the UNESCO formula (Fofonoff and Millard, 1983) and interpolated them to a common depth scale, which in our case consisted in vertical steps of 0,5 meters. The mean MHW profile is calculated then as the mean temperature value at each depth level of all the available data that concurs in time and space with the recorded event by the

GLO-REP dataset. The reference profile is the mean temperature value at each depth level of all the ARGO observations which agree in space and time of the year with each MHW singled out in 2022. Lastly, the deviation or anomaly profile is computed as the mean MHW profile minus the reference one for each event. The uncertainty for the deviation profile has been computed through a bootstrap procedure at 95% of confidence, iterating through the mean values of the MHW and reference profiles. Also, the Elzahaby et al. (2019) method allowed us to compute the mean depth of a MHW according to a

threshold calculated through the accumulated positive anomaly along the vertical dimension. The threshold modulation depends on some parametrization which in our case was chosen arbitrarily as the same that was used by the authors in order to get comparable results.

   To obtain robust results according to the available data, we decided to focus on the BSC subregion (Figure 1), given that this area contained a substantial number of ARGO profiles and MHW during the year 2022. However, data limitations arose

which implied that the long-term reference profiles were not consistent among the events, with the year of the first profile varying between 2004 and 2006 and the year of the last one between 2019 and 2021. We also had to deal with some data issues regarding fragmentation and low reliability. In this research, we discarded those profiles that were too fragmented and the specific values that were not labelled as completely reliable by the In Situ TAC.

   To analyse the subsurface daily evolution of specific MHWs we used a Hovmöller diagram of daily mean $\theta$ anomalies. This

methodology demands a dataset with regular data in time and space, and long enough to get a representative long-term reference. We achieved these requirements by using the IBI-REA from 2005 to 2021 and the IBI-NRT for 2022, calibrated as an elongation of the IBI-REA product. The calibration procedure consisted in: (i) selecting the common period for both datasets (May to December 2021) for the first 100 meters; (ii) averaging the IBI-NRT and IBI-REA $\theta$ values horizontally across the entire BSC region and interpolating both datasets to a common vertical grid of 0.5 meters, (iii) computing the

linear regression parameters of IBI-NRT to predict IBI-REA trough the ordinary least squares method (Chatterjee and Simonoff 2020, 5-8), concluding in $\beta=0.9767$, $\alpha=0.3298$, $R^2=0.990$ and significant F statistic, and (iv) correcting 2022 IBI-NRT with the regression parameters to compute the anomalies.





## 3 Results and Discussion

### 3.1 MHW characterization

The analysis of the 40-year SST time series showed that the MHWs in the IBI domain used to take place from 0 to 2 times per year, concurring with the results of Oliver et al. (2018). The annual total days take annual mean values close to 30 days; a few more days per year than the estimations of Yao et al. (2022). As shown in Figure 2, the frequency and the annual total days do not show any clear climatological zonation over the IBI domain, while for the case of the maximum intensity, it

shows a clear increment near the coastal areas reaching values of 4 °C relative to the climatology, and in relation to the duration, the maximum values of 30 days are located near the English Channel. The presence of outliers is also remarkable in some inland waters of England and Ireland, for instance, The Humber estuary (0°E, 57°N) which in small areas showed mean values of 5 MHWs events per year. Probably due to its semi-enclosed waters, which have multiple biologic, chemic and physical distinctive features (Elliott and Whitfield, 2011).

The annual mean properties from January to September 2022 indicate that the MHWs during this period were unusual. Positive anomalies are mostly found in the Celtic Sea, the English Channel and the Bay of Biscay while negative values are located in the Canary Basin and its proximities, showing a distribution that resembles the classic NAO pattern (Hurrell, 1995). Relevant negative anomalies are only found in the case of the maximum intensity with values up to -3°C around the Canary Basin. On the other hand, drastic positive anomalies appear for all the analysed parameters. The maximum intensity

property stands out by abnormal values of 2.5 °C, the duration shows local and extreme anomalies of 100 days, and the annual total days of MHW reach almost 230 days, all of them product of the events recorded per grid point on the year 2022, which in some areas had been close to 13 events (Figures 2 and 3). Taking into consideration that the year 2022 was assessed from January to September (273 days), it is important to point out that some areas did not experience MHWs only 43 days of the year.

Despite these results being, at least, quite alarming, we must point out that we understand that they may be strongly affected by the SST long term trend. As abovementioned, different authors have addressed this issue but there is not a common agreement about how to deal with SST trends and MHWs. For the IBI domain, regional studies also corroborate the influence of the SST trends on the MWHs detected in the Bay of Biscay (Izquierdo et al., 2022) and the English Channel (Simon et al., 2023). Furthermore, for the coastal areas subjected to an upwelling system such as the Canary Upwelling

System, it is considered that global warming does not produce a direct effect on MHW trends (Varela et al., 2021). In summary, we consider that Figures 2 and 3 manifest the need to establish a criterion about how to proceed with SST long term trends, because this method will be useless if all the days of the year are considered as part of a MHW.

Another way to describe 2022 anomalies was by comparing the discrete events occurred during 2022 and the record-breaking events over the past 40 years in 4 different sub-regions (Figure 1); choosing for comparisons those events which





reached the most extreme values of maximum intensity (Int. Max) and maximum duration (Dur. Max) (Table 3). From Table 2 we detected that the number of events in 2022 increased with latitude and were more intense during the summer period as also shown for previous events by Sen Gupta et al. (2020). The event of 17th May in CAN exceeded in maximum intensity the maximum duration event of 2010 in the same area by 0.28 °C; all the 2022 MHWs in IBE showed bigger absolute maximum intensity values than the maximum duration event recorded in 1997, probably due to global warming; in the BSC

area, the event starting on the 29th of April stands out for having 13 more days of duration and a greater cumulative intensity by 4.37 °C per day than the 2018 maximum intensity event; and lastly, from CEL sub-region we can highlight the event of 7th August for having 14.63 °C per day more cumulative intensity than the maximum duration event recorded in 2015-2016. Although it may not be strictly adequate to make direct comparisons between maximum duration and maximum intensity events given that intensity and duration are independent, an event can be very long and mild in intensity or vice versa. Thus,

this comparison allowed us to embrace a general perspective and observe how at least, regarding the cumulative intensity, which represents fairly well the intensity-duration interaction, two 2022 events in 2 different subregions –the 29th April event in BSC and the 7th August event in CEL– overpassed two previous record-breaking events in their respective zones.

The extreme events recorded in Table 3 allow us to link long-term physical processes with MHWs, and, consequently, with some of their impacts. Through bibliography, the influence of the NAO can be considered as one of the main drivers at least

for the cases of 2010 in CAN, and 2015 in IBE and BSC; years where Pereira et al. (2020) found the most negative (2010) and positive (2015) NAO index from 1870 to 2020. Also, described by Hu et al. (2011), the event of 2010 in CAN is even more singular as it is the longest ever registered for the IBI domain, and it is considered to be influenced not only by the negative NAO but also by the ENSO. Finally, the event recorded during June 2018 is also remarkable as it reached the highest values of maximum intensity not only for CEL but also for BSC. This event can be linked to the NAO (Simon et al.,

2023), and it is known to have had huge biological impacts in the area such as harmful phytoplankton blooms (Brown et al., 2022) or mass mortality events for mussels (Seuront et al., 2019).

### 3.2 Subsurface 2022 BSC events

The next paragraphs assess the discrete events recorded for the BSC sub-region. In Figure 4 we can observe the temperature anomaly profiles for the events detected in 2022 which featured more than 3 ARGO profiles during the MHW, and for the

maximum intensity and maximum duration events in BSC from 1982 to 2022 (Table 3), as well as the number of available ARGO profiles during the MHW and the reference period for each event, and the mean depth estimations through the Elzahaby et al. (2019) method. The anomaly profiles during the record-breaking events (grey and green profiles) show that the subsurface anomalies in BSC lie in an approximate range between -2.5 °C and 3 °C, where we ascribe the surface positive anomalies to the MHW processes, and the negative ones, appearing at depths of 30 m and below, to the ascension of

the thermocline in summer due to processes such as atmospheric blocking (Talley et al. 2011, p. 79). Other relevant results from figure 4 are: (i) the MHW mean depths calculated through the Elzahaby et al. (2019) method point out to substantial





differences between cold and warm seasonal events; MHW during cold seasons are less intense, but reach higher depths; (ii) the uncertainty inherent to the long-term reference and MHW profiles showed that subsurface interpretations had to be made carefully (iii) for the maximum duration event (green profile), we detected a drastic reduction of the uncertainty, probably

related to the higher amount of Argo observations available in this case; and (iv) the event of the 22nd of August 2022 bore strong similarities in mean anomaly profile, mean depth and also in its uncertainty ranges to the maximum intensity event for the region (grey profile).

From the Hovmöller diagram of Figure 5, obtained using IBI-REA data as long-term reference and IBI-NRT data for the 2022 MHW days, we can observe 2 MHWs, one which reaches down approximately to 10 meters and another one to 40

meters depth. Also, the formation of a layer with an intense thermal gradient of approximately 0.2 to 0.7 °C is observed, expanding from 10 to 30 meters in depth, and persisting for weeks after the end of the surface positive anomalies. This phenomenon can be understood as one of the preconditioning mechanisms for future MHWs described by Holbrook et al. (2020).

## 4 Conclusions

This study, through the usage of satellite-derived, observational and modelling data, has assessed the mean 2022 properties of the MHWs in the IBI domain, the single events in 4 different subregions and the subsurface structures of some of the events detected in the Bay of Biscay.

We showed that MHWs in the IBI domain from January 1982 to September 2022 happened on average from 0 to 2 times per

year, with a maximum mean duration of 30 days, and mean maximum intensities or deviations from the climatology of 4 °C (Figure 2). For the year 2022, the MHW frequency ranged from 0 to 16 events, with maximum mean duration values of 135 days, and mean maximum intensity values of 6 °C (Figure 3). According to the observed SST long-term trends' effect on MHWs detection by Schlegel et al. (2019) and Oliver et al. (2018), it is probably accurate to assume that these results are strongly modulated by those tendencies, meaning that we cannot ensure if extreme values are truly variating, or the MHW

temperature threshold is surpassed more often due to global warming. From the catalogue of 2022 MHWs (Table 2) we singled out two of them for overpassing record-breaking events in each sub-domain. These are the 29th April event in BSC and the 7th August event in CEL, for featuring 4.37°C·day and 14.63°C·day more cumulative intensity, an approximation to the total energy of an event, than the maximum intensity event recorded in 26th June 2018 for the BSC sub-region and the maximum duration event in CEL recorded the 19th December 2015, respectively (Table 2).


Subsurface MHW assessment in the BSC area through the ARGO dataset (Table 1) revealed a strong seasonal modulation. Cold season events reached higher mean MHW depths, around 200 meters, while the warm season ones remained shallower,



close to 20 meters; despite it is out of the scope of this study, we understand that it may be directly related with the annual variability of the mixed layer thickness, which also could explain the observed negative thermal anomalies in summer events

below 25-30 meters (Figure 4). Through model source data (Table 1) it is demonstrated how the increase of sea surface temperature, associated with the development of an MHW, is vertically moved downward in such a way that the positive anomalies persist at depth at least during weeks once the MHW has ended. In the case under investigation, the formation of a drastic thermal gradient is observed, descending from 10 to 30 meters in depth within one month (Figure 5).

**Acknowledgements**

We would like to thank Eric Oliver (Dalhousie University, Canada) for providing the main tools to detect and analyse MHWs through https://github.com/ecjoliver open source. We are also thankful to the Copernicus service for making available all kind of oceanographic data, and lastly, thanks to Miriam Selwyn (Autonomous University of Barcelona, Spain) for her wise and kind notes which have improved the text of this contribution.


**Competing interests**

The contact author has declared that none of the authors has any competing interests.

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



**Figure 1: Study area with the bathymetry, from 19ºW - 5ºE of longitude to 25ºN to 59ºN of latitude. Black boxes and its acronyms represent the areas in which we discretise the heaviest events of 2022 by spatial averaging of SST; areas are the Canary Basin (CAN) (18.5ºW - 15ºW, 30ºN - 32ºN), the Iberian Peninsula (IBE) (10ºW - 8.5ºW, 36.5ºN - 44ºN), the Bay of Biscay (BSC) (8ºW - 5ºW, 44.5ºN - 46.5ºN), and the Celtic Sea (CEL) (10ºW - 6.5ºW, 49ºN - 51ºN). This map has been obtained through Ocean Data View v.5.6.3. (Schlitzer 2021).**





**Table 1: List of Copernicus Marine products used for the computation of Marine Heat Waves (MHW) in Iberia-Biscay-Ireland region (IBI).**


| Product ref. no. | Product ID<br>Acronym<br>Type | Data access | Documentation:<br>**QUID:** Quality Information Document.<br>**PUM:** Product User Manual. |
|---|---|---|---|
| 1 | SST_GLO_SST_L4_REP_OBSERVATIONS_010_024<br>(GLO-REP)<br>Satellite observations | EU Copernicus Marine Service Product (2021) | QUID:  Good (2021)<br>PUM:  Good (2022) |
| 2 | INSITU_IBI_PHYBGCWAV_DISCRETE_MYNRT_013_033<br>(ARGO)<br>In situ observations | EU Copernicus Marine Service Product (2022a) | QUID: Wehde et al. (2022)<br>PUM: In Situ TAC Partners (2022) |
| 3 | IBI_ANALYSISFORECAST_PHY_005_001<br>(IBI-NRT)<br>Numerical models | EU Copernicus Marine Service Product (2022b) | QUID: Levier et al. (2022a)<br>PUM: Amo-Baladrón et al. (2022a) |
| 4 | IBI_MULTIYEAR_PHY_005_002<br>(IBI-REA)<br>Numerical models | EU Copernicus Marine Service Product (2022c) | QUID: Levier et al. (2022b)<br>PUM: Amo-Baladrón et al. (2022b) |






**Table 2: Record of the 2022 MHWs in the IBI area grouped by the sub-regions shown in Figure 1. The MHWs detection was applied to each sub-region using the GLO-REP product (January 1982 – September 2022). The listed events are ordered by the start date.**


|  |  | Start date | End date | Duration [days] | Intensity Max [ºC] | Cumulative Intensity [ºC·day] | Intensity Max absolute [ºC] |
|---|---|---|---|---|---|---|---|
|  | 1 | 20 Jan | 24 Jan | 5 | 0.59 | 2.75 | 11.02 |
|  | 2 | 09 Feb | 08 Mar | 28 | 0.72 | 17.60 | 10.81 |
|  | 3 | 13 Mar | 02 Apr | 21 | 0.98 | 15.47 | 10.86 |
|  | 4 | 13 Apr | 22 Apr | 10 | 1.30 | 10.43 | 11.64 |
| **C E L** | 5 | 30 Apr | 20 May | 21 | 1.95 | 34.46 | 13.21 |
|  | 6 | 26 May | 17 Jun | 23 | 2.16 | 40.06 | 15.69 |
|  | 7 | 14 Jul | 20 Jul | 7 | 1.93 | 11.92 | 18.58 |
|  | 8 | 07 Aug | 05 Sep | 30 | 3.06 | 57.42 | 20.31 |
|  | 9 | 16 Sep | 27 Sep | 12 | 1.48 | 16.07 | 17.34 |
|  | 1 | 22 Mar | 29 Mar | 8 | 0.70 | 4.92 | 12.84 |
|  | 2 | 15 Apr | 19 Apr | 5 | 1.17 | 5.33 | 13.76 |
| **B S C** | 3 | 29 Apr | 12 Jun | 45 | 2.45 | 71.89 | 16.70 |
|  | 4 | 11 Aug | 15 Aug | 5 | 1.91 | 8.10 | 21.55 |
|  | 5 | 22 Aug | 02 Sep | 12 | 1.59 | 15.49 | 21.06 |
| **I B E** | 1 | 03 Jun | 09 Jun | 7 | 1.36 | 8.83 | 18.19 |
|  | 2 | 14 Jul | 20 Jul | 7 | 1.35 | 8.10 | 19.41 |
|  | 3 | 08 Sep | 23 Sep | 16 | 2.13 | 26.04 | 20.70 |
| **C A N** | 1 | 31 Dec 2021 | 04 Jan | 5 | 0.85 | 3.97 | 20.40 |
|  | 2 | 17 May | 24 May | 8 | 1.67 | 11.26 | 21.37 |









**Table 3: List of the record-breaking MHWs grouped by the sub-regions shown in Figure 1. The first row of each group represents the strongest event in terms of maximum intensity, which is the peak point reached by the MHW relative to the climatology. The second one is the biggest event in terms of duration.**


| | | Start date | End date | Duration [days] | Intensity Max [ºC] | Cumulative Intensity [ºC·day] | Intensity Max absolute [ºC] |
|---|---|---|---|---|---|---|---|
| **C E L** | Int. Max | 26 Jun 2018 | 28 Jul 2018 | 33 | 3.86 | 86.51 | 20.30 |
| | Dur. Max | 19 Dec 2015 | 13 Feb 2016 | 57 | 0.98 | 42.69 | 11.28 |
| **B S C** | Int. Max | 28 Jun 2018 | 29 Jul 2018 | 32 | 2.76 | 67.52 | 21.34 |
| | Dur. Max | 08 Sep 2014 | 15 Nov 2014 | 69 | 2.28 | 114.99 | 18.41 |
| **I B E** | Int. Max | 04 Sep 2014 | 12 Nov 2014 | 70 | 2.65 | 139.58 | 21.11 |
| | Dur. Max | 26 Feb 1997 | 12 May 97 | 76 | 2.35 | 119.32 | 17.07 |
| **C A N** | Int. Max | 27 Jul 2004 | 10 Sep 2004 | 46 | 2.66 | 83.10 | 25.63 |
| | Dur. Max | 15 Oct 2009 | 18 Feb 2010 | 127 | 1.39 | 132.26 | 21.94 |








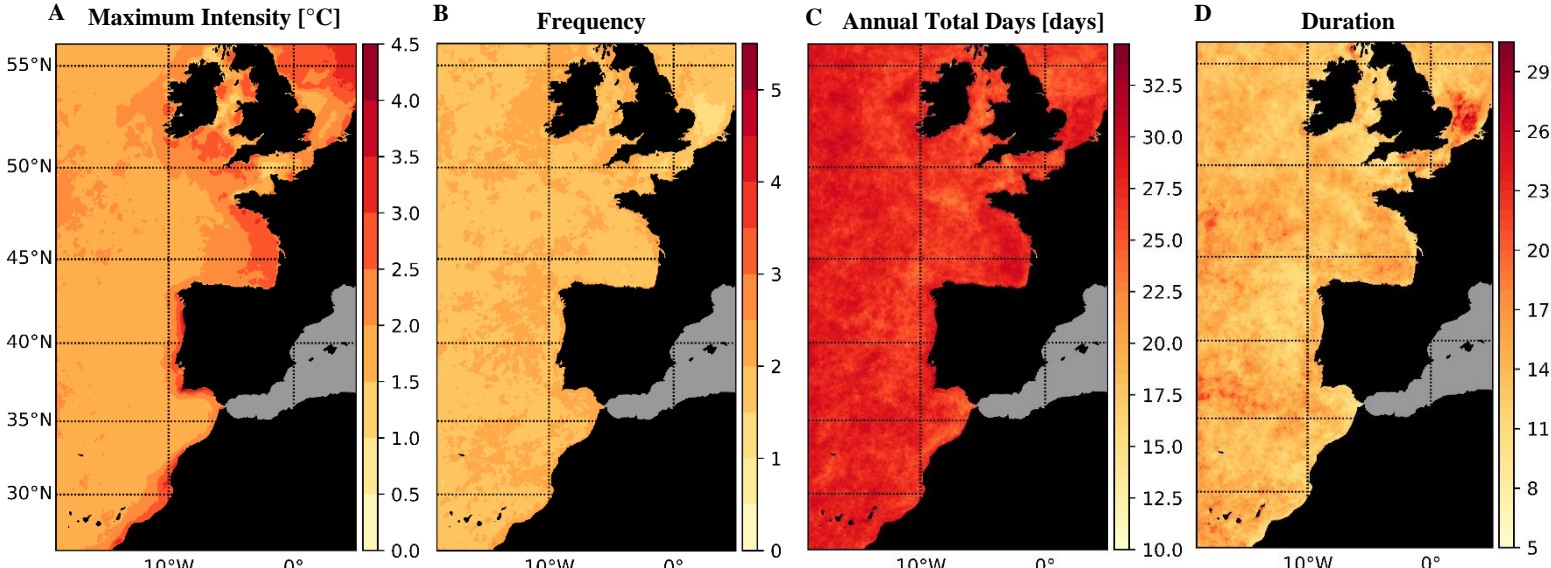

**Figure 2: Climatologic values of A) Maximum Intensity, B) Frequency, C) Annual Total Days and D) Duration, for all the recorded events through the GLO-REP dataset from January 1982 to September 2022.**






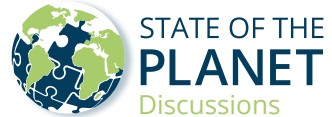







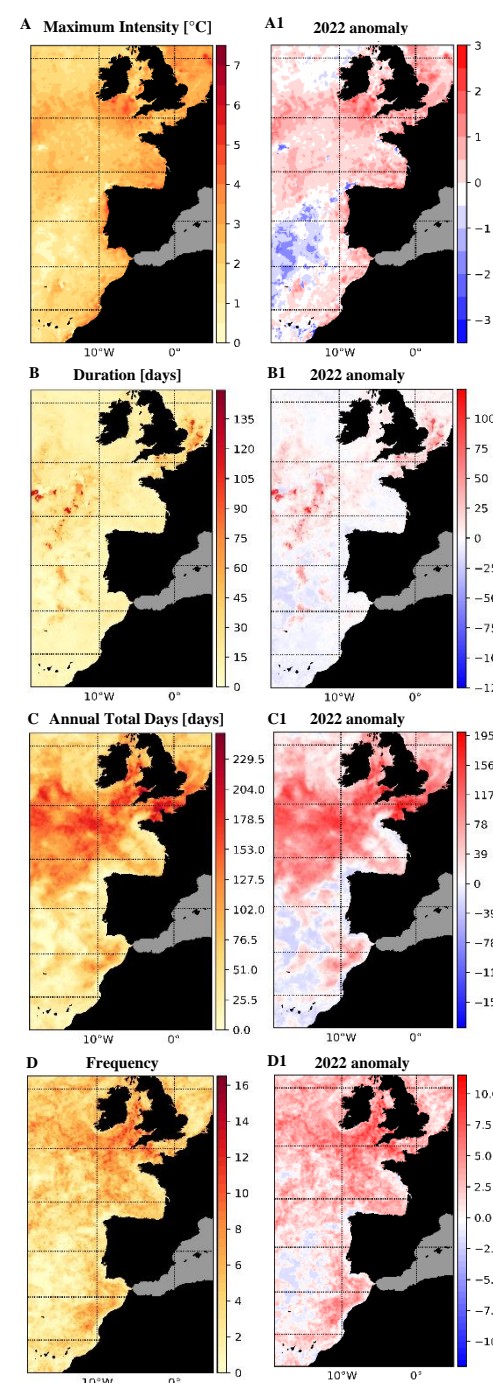

**Figure 3: 2022 mean values of A) Maximum Intensity, B) Frequency, C) Annual Total Days and D) Duration, and its respective 2022 anomaly for each parameter (A1, B1, C1 and D1). The anomaly corresponds to the 2022 mean value minus the climatologic values of Figure 2. 2022 data correspond to the GLO-REP product from January 2022 to September 2022.**



Figure 4: **Mean temperature anomaly profiles down to 350 m of depth for the BSC events with more than 3 MHW profiles and uncertainty at 95% of confidence. MHW mean depth estimation by the Elzahaby et al. (2019) method are indicated in the legend and through dotted lines. In order to facilitate the identification of each event the start date is indicated, as well as the number of profiles used in the computation of the mean profile during the MHW and the long-term reference for each event. All these results are from the ARGO dataset.**



**Figure 5: Hovmöller diagram of the mean potential temperature (θ) anomalies from 0 to 70 meters depth between the 1st of April and the 30th of June of 2022. IBI-REA is used as long-term reference from 2005 to 2021 and calibrated 2022 IBI-NRT for the MHW days. This section corresponds to a spatial average of temperature in the BSC subregion, where the dotted lines represent the start and end date of the events 2 and 3 for the BSC area recorded in Table 2. Notice that the isotherms are drawn each 0.2 ºC.**