# Peer review of "Characterization of Marine Heat Waves in the IBI Region in 2022"

_State of the Planet, 2023_

## Author Comment (AC2)

**Point by point response (Review 2):**

*Review of 'Characterization of Marine Heat Waves in the IBI Region in 2022', by Lluís Castrillo-Acuña, Axel Alonso-Valle, Álvaro de Pascual-Collar.*

Dear Anonymous referee #2,

We would like to thank you for your time dedicated into this contribution.

*The paper studies the marine heat wave event that occurred in the IBI region in 2022, using satellite data, situ observations and modelling products. The paper is clear and the problem is remarkable, but the work requires some revisions.*

*In particular, the authors analysed satellite time series of SST covering the period from 1/9/1981 to 31/09/2022, so the time series does not cover the entire year 2022, the year selected by the authors to focus their analysis. Consequently, all statistics for 2022 are distorted by the absence of the last 3 months of data. The authors should revise the text and caption of figure 3 considering the incompleteness of the time series of the 2022 SST data (9 out of 12 months) or extend the analysis to cover the entire year 2022.*

We use the data from January 1982, 1/9/1981 is the first date of available data at Copernicus website. We will extend the SST data covering the entire year 2022 having complete year data from 1982-2022.

*The authors applied the Hobday method to detect the MHWs. This method identifies the marine heat wave event whenever the SST anomaly with respect to the baseline climatology exceeds the 90th percentile threshold for more than 5 days of minimum duration. The authors state that they calculated the climatology using the entire time series of SST data instead of the 30-year climatology as suggested by the Hobday and WMO. Since the method is very sensitive to the climatology used, it is necessary to provide a justification for this choice as the base climatology and to justify the implications of using a climatology calculated using a 40-year+1 month time series.*

The baseline period for selecting the climatology has been discussed in the OSR coordination meetings. The conclusions regarding this issue for all the groups that will assess MHWs in the OSR8 where:

1. Each group can choose the referee period.
2. The period should contain the year 2021.
3. The period must be the same for all datasets used to compute MHW in a contribution.

Hobday et al. (2016) recommends a 30 -year length referring to the WMO; so, there is just one source for justify the 30 -year length, the WMO. The Guide to Climatological Practices (WMO-No. 100) page 75 said:

*"The 30-year period of reference was set as a standard mainly because only 30 years of data were available for summarization when the recommendation was first made."*

So, the 30 -year period seemed to be a minimum recommended length instead of a specific length. There are also standardized periods to calculate the climatology, updated each decade as used in the study that you mentioned. But as we are suggested to include the year 2021 and the last WMO references period covers from 1991-2020, the standard periods are not an option.

We also perform an internal test to see into how sensitive our results regarding this issue are. Here we show the frequency case; the climatological values, the count for the year 2022 and the 2022 anomaly (both cases with the updated dataset including all the year 2022). The 2022 count and anomaly perform indistinctly according to all the region, which are the core of our study. Some differences may be seen in the normal values despite they are not much relevant. In any case, any of our conclusions change.

[Figure]

As it is not a trivial issue, we will add some justification for the usage of all the period on computing the climatology instead of just 30 -years.

*A recent paper (Marullo et 2023) analysed the exceptional MHW of 2022 in the Mediterranean Sea, demonstrating that the 2022 MHW event persisted during the autumn of 2022 and early winter of 2023. The authors demonstrated that the occurrence and growth of sea surface temperature anomalies is linked to the prevalence of a persistent anticyclonic system over the Mediterranean area. The article by Marullo et al. should be cited and the extension of the IBI analysis to the entire 2022 SST time series is strongly recommended. In case the extension of the analysis to the entire 2022 time series is not possible, the IBI MHW result should at least be discussed taking into account the results of Marullo et al. 2023.*

Fortunately, we will have the 2022 data for our region, and we will not have to extrapolate results from other regions. On the other side, looking in the new events recorded through the data extension this recommended paper will be used to discuss them.

*Finally, the discussion on the removal of the SST trend before identifying MHW and the effect this has on the results obtained should be analysed and discussed more carefully.*

This contribution do not has the objective on assessing the magnitude of the influence of the long-term SST trends has on the MHWs results. We applied the standardized procedure to compute MHWs, which says nothing about detrending the data. In this contribution, applying the standard methodologies we obtain very abnormal results where we believe that they could be conditioned by the SST long term trend. In this way, we consider that this issue should be more discussed in the community because, if not, it seems that the standard method may become useless in the framework of a climate change scenario. In the work, we deliberately avoid this discussion, because it is something out of the scope of this contribution, and also because of the length of the contribution. However, given that the influence of trends on the methodology is currently a subject under discussion, we must inform the readers of the potential impact that long-term trends may exert on the results.

Nevertheless, we propose modifying the manuscript by providing additional information on the potential consequences of trends in calculating MHW as described in the current literature for the region.

*The analysis of MHW propagation in depth using the ARGO profile should be better described and analysed. The anomaly of the SST profiles for the days before the start of the MHW events should be shown and compared with the SST anomaly during the events. This will make it possible to identify whether, for example, the anomaly at depth is due to vertical propagation of surface warming or whether the SST anomaly observed at depth is due to advection due to changes of circulation.*

We all agree that this is a good point. Unfortunately, data limitations in this case are relevant. ARGO data is an irregular sample in time and space increasing drastically the uncertainty of the results. According to the available data we consider that this kind of assessments may be adventurous.

*Figure 2. Colours are fully saturated. The colours should be improved. The climatology should be calculated using whole years (January 1982 to December 2021) instead of January 1982 to September 2022.*

We will take care of it.

*Figure 3. Since the analysis does not cover data for the entire year 2022, the annual average values do not make sense. The authors should provide information on how the annual values are calculated.*

The analysis will cover the entire 2022.

---

## Author Response (AR1)

**Point by point response (Review 1):**

Review of "**Characterization of Marine Heat Waves in the IBI Region in 2022**", by Lluís Castrillo-Acuña, Axel Alonso-Valle, Álvaro de Pascual-Collar.

*The article details MHW events occurring at the large IBI region, using data from 1981 to 2022 from various sources (satellite reanalysis, models and in situ data for the 2022 events). The paper reads well and is well organized. I have two major comments, which could be summarized as follows:*

*- Q:Some analysis are presented for 2022 (like in figure 3) and the text that refers to it, but the satellite data only covers until September of that year. I think it can be misleading to discuss about variables like maximum intensity, annual total days and so on, when the statistics do not cover the whole year. As it's necessary for these data to become available, I would encourage the authors to limit this kind of analysis and instead go more in depth with specific events that occurred during that year. This takes me to my second major comment:*

R: You are completely right; we all agree that this is probably the major issue in this contribution. In this way we have been able to update our dataset covering all the year 2022, consequently, the affected figures and text have been corrected in the main document.

Changes in line 14, 103, 154, 162, 194, 265. Figure 2 and 3. Table 2 and 3.

*-Q: The authors present some analysis of the effect of the MHWs at depth, but in my opinion these results are not fully exploited. In situ profiles are only presented for one region, and the figure (figure 4) is quite crammed and therefore unclear. I would expand this part of the paper to provide more insight into the effects of MHWs at depth in different regions, as this is something that needs more insight in order to understand the effects of MHWs in the ecosystem.*

R: According to the expected contribution length and the fragile robustness of the results looking into the data limitations, we consider that we cannot (despite it is truly interesting) expand this analysis (figure 4) until there is more *in situ* subsurface data. We also consider that expanding the conclusions regarding these specific results could be too adventurous, as one of our main conclusions is that subsurface MHWs assessment through ARGO data is currently strongly limited (which we try to demonstrate by putting the confidence intervals of the 4 MHWs events with more available ARGO data).

To expand subsurface assessment, we tried to ugrade the subsurface results through Figure 5 (the Hovmöller diagram). Changes from line 238.

*Detailed comments;*

*Q: line 20: which perform extreme temperatures -> which show/result in*
R: Applied. Change in line 20

*Q: line 22: the authors mention the global temperature trend but not the fact that because of it, the number of MHWs and their intensity is rising, which I guess is the point they want to make here?*
R: You are right. We added a clarification at the end of the sentence. Change in line 23

*Q: line 33: again the authors say something but don't explain it fully, which for non-experts might be confusing. The discussion on removing the SST trend before detecting MHWs and the effect this has on the results obtained should be detailed more carefully.*
R: We reordered an added a sentence trying to clarify this point. Change from line 35

*Q: line 141: (5-8): not sure what these numbers refer to.*
R: It is an error. Corrected. Change in line 148

*Q: line 146: "used to take place" when?*
R: We added a clarification on the text. Change in line 153

*Q: line 151: outliers in inland waters. Why are inland waters, not addressed in this work, not masked out? Are these really outliers? and are they removed from the analysis before MHW assessment? (if yes, I would take them out of the figures, if not, they might be influencing the results which is not good)*

*R:*

(i)  *Why are inland waters, not addressed in this work, not masked out?:* We avoided this terminology in order to simplify and not lead to misunderstandings. Change in line 159.

(ii)  *Are these really outliers?* To be more robust in our writing we changed from outlier to abnormal value. Change in line 159

(iii)  *and are they removed from the analysis before MHW assessment? (if yes, I would take them out of the figures, if not, they might be influencing the results which is not good)* We do not remove anything before neither after the MHW assessment.

*Q: line 153: "This is" probably due to (otherwise sentence non complete)*
R: Corrected. Change in line 160

*Q: line 155: refer to figure 3 in the first sentence of this paragraph.*
R: Done. Change in line 163

*Q:  line 156: "positive anomalies": of what. And later in this line, "negative values": of what?*

*Q: line 159: "Drastic" positive anomalies. Define drastic, or use another, more quantifiable word*

R: Corrected. Drastic to severe. Change from line 163

*Q: line 163, I would turn the last sentence the other way round: indicate the number of days of MHW, as you have just given the number of days of the analysed time series.*

R: We will removed this sentence.

*Q: line 183: incomplete sentence*

R: Sentence expanded. Change in line 202

*Q: line 207: "cold and warm seasonal events" -> events during cold and warm seasons.*

R: Corrected. Change in line 232

*Q: Figure 2. The colors are completely saturated, and there is no reason to start the minimum values where they stand now (minimum intensity of MHW at 0degC?, Annual total days at 10 days?) It would improve the clarity of the figure if these were adjusted. Also decreasing the number of colours in panels C and D, as it is done in panels A and B. It is not possible to differentiate between 25 and 32 days in panel C for example.*

R: We completely agree with this comment. We worked to find a better way of representation in order to minimise losing information and maximise the visualization. Changes in Figure 2 and 3.

*Q: Figure 3. I am not sure highlighting 2022, for which there is not a complete year of data available in this paper, is a good idea. Wouldn't it make more sense to wait until the whole 2022 is available (it isn't yet?) so that numbers can be compared to other years?*

R: Dataset expanded.

*Q: Figure 4. Mention in the caption that these are 2022 data. A small insert with the position of these profiles would be interesting. Why only profiles in one region are shown? It would be interesting to compare with what happened at the more coastal IBE region, or the shallow CEL region... As I mentioned above, figure 3 is not complete as the data used do not cover the whole year, therefore the authors could choose to leave maybe this figure out (and put average numbers in a table for example?) and use the extra figure to*

*look more in detail at the depth variations in different regions.*

R: We added a small insert with the position of the ARGO profiles used to obtain the figure 4 results in figure 1. Change in figure 1

We only select one region to perform the subsurface assessment for different reasons. First, this contribution is expected to be relatively short in comparison to a common paper, so, doing this analysis for all the subregions could be probably enough to a single contribution. Second, there are substantial data limitations. We chose the BSC region to be the one with more quantity of *in situ* data per MHW event, even thought, as shown in figure 4, the confidence intervals are too wide, even crossing to 0 sometimes. Meaning that conclusions by using this data must be conservative.

*Q: Figure 5. Only one paragraph is dedicated to this figure, which is surprising. Authors should add perhaps a time series of the intensity of the MHW at the surface, and indicate up to which depth these anomalies are still considered a MHW. Since these come from a model, looking at water currents at different depths could be also of interest, or wind intensity… Compare with other regions... There is a need to assess what the impact of the MHWs is in the subsurface and this work could help improving our knowledge on that.*

R: We added a time series from the truly MHWs result, the surface assessment through the L4 reprocessed data. Also, we discussed about different topics opened by these results.

Changes from line 238 and figure 5

**Point by point response (Review 2):**

*Review of 'Characterization of Marine Heat Waves in the IBI Region in 2022', by Lluís Castrillo-Acuña, Axel Alonso-Valle, Álvaro de Pascual-Collar.*

*The paper studies the marine heat wave event that occurred in the IBI region in 2022, using satellite data, situ observations and modelling products. The paper is clear and the problem is remarkable, but the work requires some revisions.*

*In particular, the authors analysed satellite time series of SST covering the period from 1/9/1981 to 31/09/2022, so the time series does not cover the entire year 2022, the year selected by the authors to focus their analysis. Consequently, all statistics for 2022 are distorted by the absence of the last 3 months of data. The authors should revise the text and caption of figure 3 considering the incompleteness of the time series of the 2022 SST data (9 out of 12 months) or extend the analysis to cover the entire year 2022.*

We extended the SST data covering the entire year 2022 having complete year data from 1982-2022.

Changes in line 14, 103, 154, 162, 194, 265. Figure 2 and 3. Table 2 and 3.

*The authors applied the Hobday method to detect the MHWs. This method identifies the marine heat wave event whenever the SST anomaly with respect to the baseline climatology exceeds the 90th percentile threshold for more than 5 days of minimum duration. The authors state that they calculated the climatology using the entire time series of SST data instead of the 30-year climatology as suggested by the Hobday and WMO. Since the method is very sensitive to the climatology used, it is necessary to provide a justification for this choice as the base climatology and to justify the implications of using a climatology calculated using a 40-year+1 month time series.*

We justified the usage on using all the period because in this way we can use more values on computing mean values and because any period selection would by arbitrary.

Change in line 106.

*A recent paper (Marullo et 2023) analysed the exceptional MHW of 2022 in the Mediterranean Sea, demonstrating that the 2022 MHW event persisted during the autumn of 2022 and early winter of 2023. The authors demonstrated that the occurrence and growth of sea surface temperature anomalies is linked to the prevalence of a persistent anticyclonic system over the Mediterranean area. The article by Marullo et al. should be cited and the extension of the IBI analysis to the entire 2022 SST time series is strongly recommended. In case the extension of the analysis to the entire 2022 time series is not possible, the IBI MHW result should at least be discussed taking into account the results of Marullo et al. 2023.*

Cited in line 208

*Finally, the discussion on the removal of the SST trend before identifying MHW and the effect this has on the results obtained should be analysed and discussed more carefully.*

We provided additional information on the potential consequences of trends in calculating MHW as described in the current literature for the region.

Change line 183

*The analysis of MHW propagation in depth using the ARGO profile should be better described and analysed. The anomaly of the SST profiles for the days before the start of the MHW events should be shown and compared with the SST anomaly during the events. This will make it possible to identify whether, for example, the anomaly at depth is due to vertical propagation of surface warming or whether the SST anomaly observed at depth is due to advection due to changes of circulation.*

We all agree that this is a good point. Unfortunately, data limitations in this case are relevant. ARGO data is an irregular sample in time and space increasing drastically the uncertainty of the results. According to the available data we consider that this kind of assessments may be adventurous.

*Figure 2. Colours are fully saturated. The colours should be improved. The climatology should be calculated using whole years (January 1982 to December 2021) instead of January 1982 to September 2022.*

Climatology calculated using whole years and figure 2 and 3 changed.

*Figure 3. Since the analysis does not cover data for the entire year 2022, the annual average values do not make sense. The authors should provide information on how the annual values are calculated.*

The analysis covers the entire 2022.